# Epigenetics and the Tumor Microenvironment in Neuroendocrine Tumors

**DOI:** 10.3390/cancers18010069

**Published:** 2025-12-25

**Authors:** Alice Castenetto, Teresa Gagliano

**Affiliations:** Cancer Cell Signalling Lab, Department of Medicine, University of Udine, 33100 Udine, Italy; castenetto.alice@spes.uniud.it

**Keywords:** neuroendocrine neoplasms, epigenetics, crosstalk, tumor microenvironment, targets, therapeutic strategies

## Abstract

The incidence of neuroendocrine neoplasms is increasing over the years, and therapy effectiveness is still relatively limited by the development of resistance mechanisms. Given the low rate mutation of these neoplasia, there is evidence of an emerging role of epigenetic modifications in neuroendocrine tumor pathogenesis and therapy resistance, leading to a need to investigate more deeply how these mechanisms are altered in these tumor types, how they can affect therapeutic efficacy, and if they can mediate the crosstalk between tumor cells, tumor microenvironment components, and tumor-stroma crosstalk that support tumoral development and progression. In this review we aim to describe all the recent evidence involving epigenetic mechanisms and the tumor microenvironment components that support tumor onset, development and resistance to treatment in the context of neuroendocrine neoplasm. The goal is to highlight potential new path to improve treatment response and outcomes in patients affected by neuroendocrine neoplasms.

## 1. Neuroendocrine Neoplasms

Neuroendocrine neoplasms (NENs) represent a group of rare epithelial malignancies, with a heterogeneous spectrum that ranges from well-differentiated neuroendocrine tumors (NETs) (80–90%) to very aggressive poorly differentiated neuroendocrine carcinomas (NECs) (10–20%) Table 1 [1]. These neoplasms arise from neuroendocrine cells diffused throughout the body, although the most frequent locations are the gastroenteropancreatic (GEP) and bronchopulmonary tracts [2], and are associated with symptoms secondary to the release of vasoactive amines and hormones [3]. NENs are mostly sporadic, but in 10–30%, they can arise within the context of inherited genetic syndromes, such as MEN1 (multiple endocrine neoplasia type 1), NF1 (neurofibromatosis type 1), TSC1/2 (tuberous sclerosis), and VHL (Von Hippel–Lindau disease) [4].

## 2. Epigenetics of NETs

Epigenetics represent the heritable and stable changes in gene expression determined by chemical modifications of DNA and its associated proteins, rather than alterations to the DNA sequence [12]. The most common epigenetic mechanisms include DNA methylation status, histone post-translational modifications, and regulation by noncoding RNAs (ncRNAs) [13] Figure 1.

Neuroendocrine tumors present a low mutation rate compared to other tumors, while the role of epigenetic mechanisms regulating gene expression seems to be relevant in their pathogenesis and development; this supports the rational to further investigate the epigenetic mechanisms at the basis of NEN pathogenesis. Furthermore, epigenetic modifications are reversible and targetable, representing an attractive target for treatment [14], assuming prior clarification of their molecular role in NET pathogenesis.

At present, the gold standard for NET treatment is surgical removal of the tumoral mass, while new methods are being studied, such as preoperative contrast-enhanced computed tomography (CE-CT), to implement tumor characterization [15]. Whenever, in more aggressive NETs, surgery is not feasible, somatostatin analogs (SSAs), chemotherapy, targeted therapies, and specialized radiation, such as peptide receptor radionuclide therapy (PRRT), could be used, despite often leading to resistance development.

This approach may also allow, through combination with established medical therapies—including classical somatostatin analogs (SSAs), targeted agents, and cytotoxic chemotherapy—the overcoming of resistance mechanisms, thereby achieving more effective treatment strategies for neuroendocrine tumors [16,17]. In more aggressive NETs, for which surgical intervention is not feasible, SSAs are currently used; however, their efficacy is often limited by the development of therapeutic resistance. The proposed causative mechanisms for development of resistance upon SSA administration include epigenetic changes within tumor cells, as histone methylation and/or acetylation, that reduce SSTR expression. Similarly, since high SSTR2 levels are also essential for PRRT to ensure efficient delivery of radiation, epigenetic modifications that reduce SSTR expression can lead to resistance development. This hypothesis is supported by the observation that using HDAC inhibitors (HDACi) and DNMT inhibitors (DNMTi), NET cells presented an overexpression of SSTR2 [13]. Consequently, the combination of HDACi and/or DNMTi with SSAs or PRRT might overcome this epigenetically derived resistance, resulting in a more efficient antitumoral treatment [18]. This combination is also attractive for targeted therapies, including mTOR or VEGFR inhibitors, such as Everolimus and Sunitinib, respectively. In fact, the potential involvement of epigenetic mechanisms in Everolimus resistance development is supported by the observed dysregulated expression of the downstream mTOR components [19]. Similarly, Sunitinib is limited in its efficacy by the development of resistance mechanisms that are in part epigenetically derived, such as the activation of HIF-1α that in turn is responsible for the activation of alternative pathways associated with the expression of pro-angiogenic factors [20].

Given that neuroendocrine tumors present a low mutational burden, it is fundamental to identify new epigenetic markers rather than genetic ones, for both diagnosis and prognosis, which would also allow a more precise distinction between different subtypes, which can be treated with more personalized and efficient therapies. Recently, different epigenetic markers are under investigation in neuroendocrine tumors. For instance, alterations in methylated levels of ZNF536, encoding for a transcriptional repressor, have been proposed to have both diagnostic and prognostic implications in NETs [21]. Furthermore, the expression of mitochondrial genes related to oxidative stress (MTGs-OS) that is epigenetically regulated can be considered a prognostic marker that also provides clinical guidance for PNET treatment [22]. There are even some biomarkers reflecting the interaction between tumor cells and the tumor microenvironment, which are relevant to tumor progression and metastasis, such as the sEV-miR-183-5p prognostic marker, a noncoding RNA embedded in small extracellular vesicles promoting the polarization of tumor-associated macrophages into the immunosuppressive M2 phenotype and upregulating SPP1, which represents a good candidate for a therapeutic target [23].

### 2.1. DNA Methylation

DNA methylation is catalyzed by DNA methyltransferases (DNMTs) [24] and frequently occurs on cytosine residues that precede guanine nucleotides or CpG sites, with the exception of CpG islands [25]. Perturbation of methylation patterns could give rise to gene expression alterations responsible for malignant cell transformation [26]. In particular, hypermethylation of promoter regions in tumor suppressor genes often results in their silencing, contributing to tumor growth and progression [27].

In lung NETs, the hypermethylation of many tumor suppressors has been reported, such as the cell-cycle inhibitor CDKN2A [7,28], RASSF1A, a scaffold protein involved in cell signaling pathways such as Erk, Hippo, apoptotic, and p53 [29,30], SOX17, a transcription factor inhibiting proliferation of lung cancer cells and downregulating the Wnt/β-catenin pathway [31], and TAC1, which encodes for tachykinins, involved in neurogenic inflammatory processes and in endotoxin-induced airway inflammation (Figure 2A) [32,33].

Similarly to pulmonary NETs, the majority of pancreatic NETs present hypermethylated *RASSF1A* and *CDKN2A* genes [34]. Other hypermethylated tumor suppressors in PanNETs include *PHLDA3*, which downregulates the PI3K/AKT oncogenic signaling pathway [35], *SSTR2*, a G-protein coupled receptor that inhibits the release of hormones [36], *APC*, an antagonist of the Wnt/β-catenin pathway, and *MEN1* and *MGMT*, encoding for DNA repair proteins involved in cellular defense (Figure 2B) [37,38]. Moreover, many PanNETs are characterized by significant hypomethylation of *LINE-1* and *Alu* sequences [39], resulting in transcriptional activation and consequently in genetical instability [40].

In gastrointestinal NETs, promoters of *RASSF1A*, *SEMA3F*, encoding for semaphorin 3 that induces apoptosis, and *CTNNB1*, representing the catenin β1 gene that constitutes adherence junctions to regulate cell growth and inter-cellular adhesions, are hypermethylated [41]. Moreover, *MAPK4*, which encodes for a kinase regulating proliferation, migration, and apoptosis via the AKT/mTOR and/or PDK1 signaling pathways [42,43], and *WIF1*, which is a Wnt antagonist, are silenced through hypermethylation (Figure 2C) [44]. Similarly to what is reported for PanNETs, in many GI-NETs, *LINE1* and *Alu* repetitive elements are also hypomethylated, causing genetical instability [42,45].

MTC, which represents the most common thyroid NET subtype, presents some genes regulated in their expression by diverse methylation levels, such as *MGMT*, which has been shown to be hypermethylated at the level of the CG_16698623 dinucleotide [46], *RASSF1A*, which is frequently hypermethylated in thyroid NETs as a promoter of endocrine-related genes, including *SLC5A5* and *THST*, and as *CASP8* and *ATM* genes (Figure 2D). It is interesting to note that nondifferentiated medullary tumors exhibit more hypomethylated than hypermethylated genes. Some examples of hypomethylated genes in MTC are *INSL4* and *DPPA2*, which represent two genes reported to promote tumorigenesis in different cancer types, acting as oncogenes [47].

### 2.2. Histone Modifications

Histone modifications, such as methylation, acetylation, and phosphorylation of amino acidic residues on histone protein tails, affect chromatin structure and accessibility, regulating gene expression [48]. They represent an important epigenetic mechanism to regulate the chromatin state and consequently gene expression. If altered, some of them can contribute to the pathogenesis of tumors, including neuroendocrine neoplasms [7]. Acetylation levels are regulated by the opposite activity of HATs (histone acetyltransferases) and HDACs (histone deacetylases), which have been shown to be altered in different cancer types [26]. HDACs have been under investigation as potential drivers of cancer progression since their inhibition has been shown to reverse the malignant phenotype in various tumor entities [49], including NETs [17] (Figure 1).

In pulmonary NETs, a significant correlation has been reported between the acetylation level of histone H3 surrounding the promoter region of *Notch1* gene and its expression. In fact, histone deacetylation results in *Notch1* silencing in small-cell lung cancer (SCLC) (Figure 2A) [50]. Expression of *Notch1* in SCLC is associated with a reduction in its proliferation while promoting cell apoptosis, a reduction in cell motility, invasion, and metastasis, and enhancement of cell–cell adhesion by EMT inhibition [51]. Compared to histone H3, H4 acetylation has a wider distribution along transcribed genes, and there is evidence that histone H4 acetylation at lysine 16 (H4KA16) and trimethylation at lysine 20 (H4KM20) gradually decrease from low- to high-grade in lung NETs [52]. KMT2 proteins regulate methylation of lysine 4 and 27 of histone 3 tails (H3K4 and H3K27). In SCLC, mutations in *KMT2* genes are associated with low protein levels, resulting in a reduction in mono-methylation of histone H3 lysine 4 (H3K4me1) [53] and in histone and DNA hypomethylation, finally affecting *DNMT3A* expression levels [54].

In PanNETs, loss of menin is responsible for H3K4me3 loss, because menin represents a subunit of an MLL1/MLL2-containing histone methyltransferase complex that tri-methylates histone H3 at lysine 4, enhancing transcriptional activity [55]. Thus, *MEN1* loss alters the epigenetic landscape of its target genes, such as *IGF2*, *CDKN2C*, and *CDKN1B* (Figure 2B) [56]. Furthermore, in some PanNETs, downregulation of the *WIF1*, *DKK-1*, and *DKK-3* genes has been observed, which is caused by repressive histone modifications resulting in an increased H3K9me2 presence at the promoter level [38].

In primary intestinal NETs, a high expression of di-methylation on lysine 4 of histone 3 (H3K4) has been observed [53]. Unfortunately, despite evidence of DNA methylation levels regulating SST2 expression, there are still some gaps related to an effective regulatory role of histone modifications, maybe due to the limited sample size. However, the effects of HDACi on *SST2* expression led researchers to think about a possible role of histone acetylation in SST2 expression regulation in pancreatic NETs (Figure 2C) [57,58].

Concerning thyroid NETs, in more aggressive MTCs, a significant increase in the EZH2 and *SMYD3* genes has been observed. An *EZH2* increase leads to the silencing of genes that promote differentiation and restrain proliferation and to altered regulation of genes involved in the Wnt/β-catenin, Erk, and Akt signaling pathways, which are mainly involved in MTC tumorigenesis (Figure 2D). SMYD3 is a histone methyltransferase that regulates several target genes, including cell cycle mediators and oncogenes [59].

### 2.3. Noncoding RNAs

MicroRNAs (miRNAs) and long noncoding RNAs (lncRNAs) regulate gene expression at the post-transcriptional level [60,61]. In more detail, miRNAs are involved in RNA interference machinery suppressing the protein translation or decay of mRNA by binding to the untranslated regions (UTRs) of mRNA. Through this mechanism, miRNAs act as epigenetic modulators by targeting enzymes involved in epigenetic reactions and, in turn, miRNA expression is regulated by epigenetic machinery, forming the miRNA–epigenetic feedback loop [62,63]. Long noncoding RNAs mediate DNA and histone modifications and chromatin organization by recruiting DNA- and histone-modifying enzymes and chromatin organization-associated proteins at specific genomic sites to regulate pre-transcriptional gene expression [64].

Noncoding RNAs are involved in the regulation of proliferation, differentiation, metabolism, and cell death, and their dysregulation is implicated in oncogenesis, as observed in NETs [65].

Consistently, lung NETs present an altered ncRNA pattern, including upregulation of miR-21, which directs cell growth by inhibiting apoptosis, and miR-34a, which acts as a tumor suppressor (Figure 2A) [66]. Many other ncRNAs have been shown to be altered in their expression, supporting tumor pathogenesis, such as miR-1 downregulation, which is associated with SCLC growth and metastasis through the regulation of the CXCR4/FOXM1/RRM2 axis [67], and lncRNA PROX1-AS1 upregulation, which acts as a tumor promoter [68].

Additionally, in pancreatic NETs, the upregulation of miR-21 has been observed, similarly to what has been observed for miR-196a, which regulates Akt signaling, and miR-670-3p, which regulates claudins CLDN1 and CLDN2, whose dysregulated expression is associated with tumor initiation, progression, and metastasis (Figure 2B) [69,70,71]. In addition, miRNAs could be used by tumors as a mechanism of adaptation to hypoxia-related stress, as demonstrated by upregulation of miR-210 induced by HIF-1α [70]. PanNETs also present alterations in lncRNA expression patterns, which can contribute to tumor pathogenesis, as observed for the overexpression of lncRNA XLOC_221242 compared to normal tissue, which is positively related with DNER mRNA, a factor involved in the Notch signaling pathway [72]. Another example involves *MEN1*, which, in physiological conditions, activates lncRNA MEG3, which downregulates *c-Met* proto-oncogene expression, acting as a tumor suppressor. In *MEN1*-mutated PanNETs, since MEG3 is not activated, it does not present a *c-Met* reduction [73].

Many dysregulated ncRNAs were also identified in GI-NETs, whose altered expression is linked to different mechanisms that support tumor development. For instance, in rectal NEN upregulation of miRNAs, including miR-885-5p, miR-135a, miR-198, miR-204, miR-216a/b, miR-452, miR-486-3p, miR-499, and miR-146b-3p, has been associated with the invasion of lymphatic vessels. Furthermore, miR-186 downregulation in colorectal NETs is related to *PTTG1* overexpression, resulting in the promotion of cell proliferation, invasion, and metastasis [74,75]. In addition, dysregulated lncRNAs were also identified in GI-NETs, such as HOTAIR, which acts as a chromatin state regulator, which could represent a potential therapeutic target against cancer (Figure 2C) [76].

In both familial and sporadic MTC forms, dysregulated expression of ncRNAs has been observed, such as overexpressed miR-375, which targets a growth inhibitor gene, *YAP1*, which is a transcription factor involved in the Hippo signaling pathway, which presents a role in development, growth, repair, and homeostasis (Figure 2D) [77].

### 2.4. Emerging Epigenetic Mechanisms

As well as the previously described more “classical” epigenetic processes, others have recently been discovered to contribute to neuroendocrine tumor biology, such as RNA methylation and chromatin-remodeling complexes (Figure 1).

The RNA methylation pattern is a critical post-transcriptional regulator mechanism that governs gene expression. As confirmed by many tumor types, aberrant N^6^-methyladenosine (m^6^A) modification of ncRNAs may affect cancer progression [78]. Consistently, in sparsely granulated subtypes of pituitary neuroendocrine tumors, the fat mass and obesity-associated protein (FTO), an m6A demethylase, was observed to be upregulated and, in turn, modulate the m6A levels of the mRNA encoding for DSP, a critical desmosomal component. This results in a reduction in desmosome organization, leading to enhanced tumor invasiveness and metastasis [79].

Another recently highlighted epigenetic mechanism that plays an important role in NET pathogenesis involves the Switch/Sucrose Nonfermentable (SWI/SNF) complex, a chromatin remodeling complex that regulates nucleosome structures. In particular, in NETs, the aberrant expression of SWI/SNF complex subunits affects downstream gene expression, including NE markers, highlighting its emerging role in NET pathogenesis [80].

## 3. Tumor Microenvironments in NETs

The tumor microenvironment (TME) represents a complex and dynamic surrounding that actively interacts with tumor cells, affecting processes such as cancer development, progression, metastasis, and immune evasion. Although its specific composition varies between different tumor types, the TME includes immune cells of innate and adaptive immunity, stromal cells, extracellular matrix (ECM), and blood vessels [81].

Immune cells, including B and T cells, NK cells, mast cells, dendritic cells, and macrophages, are known to infiltrate NETs, creating an immunosuppressive microenvironment that enables tumor progression [82]. In recent years, there has been evidence of the role of TME and tumor-infiltrating lymphocytes (TILs) in cancer prognosis and therapy response. In this regard, a study on neuroendocrine tumors of the lung demonstrated that in LCNECs, high levels of CD8+ TILs and PD-L1 expression may reflect a greater responsiveness to immune checkpoint inhibitors compared to carcinoids, characterized by a distinct TME with a desert-immune phenotype, which may show a limited benefit from immunotherapy. Consequently, an integrated approach combining molecular profiling and immune data for different neuroendocrine tumor types may define a more effective treatment [83].

ECM degradation contributes to NET development and progression by releasing chemoattractants that recruit inflammatory cells and pro-angiogenic factors and creating a mechanic stimulus that may influence tumor behavior [82].

Among stromal cells, fibroblasts present a crosstalk with NET cells, where tumor cells stimulate the proliferation and activation of fibroblasts by secreting soluble factors, including TGF-β, serotonin, and PDGF, and, in turn, activated CAFs modulate the proliferative capability of NET cells. Besides soluble factors, the means of communication between TME components, such as CAFs, and tumor cells are almost unknown, although epigenetic modifications represent plausible candidates. Supporting this concept, emerging evidence indicates that epigenetic regulation of tumor cells by the TME occurs in neuroendocrine tumors. In particular, hypoxia within the TME of gastroenteropancreatic NETs (GEP-NETs) has been shown to induce promoter hypermethylation-mediated downregulation of carbonyl reductase 4 (CBR4), thereby promoting tumor progression through an ubiquitin-dependent reduction in fatty acid synthase (FASN) expression and the activation of the mTOR signaling pathway [84].

As previously mentioned, there is still limited information reported in the literature about specific mechanisms that TME elements use to regulate epigenetic modifications in NET cells. Nevertheless, some hypotheses, which, in future, may be interesting to investigate in NET cells, can be proposed based on the non-NE tumor knowledge on this topic.

In general, cellular components of the TME, such as TAMs, MDSCs, DCs, and CAFs, secrete different cytokines, among which members of the IL-6 family, including leukemia inducible factor (LIF), IL-10, TNF-α, and TGF-β [85], that affect both tumor cells and other stromal components of the TME through different mechanisms, including epigenetic regulation. For instance, transient TGF-β stimulus can induce LIF expression in both tumor and stromal cells via DNA hypomethylation of CpG pairs and changes in histone methylation status [86].

In different tumor types, the epigenetic modifications induced by CAFs in tumor cells have been analyzed, such as in ovarian cancer cells, where it was observed that CAFs prompt EZH2 histone methyltransferase upregulation, which results in cancer cell migration and increased invasion [87]. Similarly in SI-NETs, a high differential expression of EZH2 has been reported; this catalyzes H3K27me3 marks and is associated with high proliferation rates, metastasis, and poor survival [34]. Hence, we can hypothesize that neuroendocrine tumor cells may also be influenced by CAFs present in the TME through epigenetic regulation, as suggested by the EZH2 upregulation observed in SI-NETs and thyroid NETs.

The crosstalk between CAFs and tumor cells was further confirmed by the overexpression of TNF-α observed in ovarian cancer, which is determined by promoter DNA hypomethylation and chromatin remodeling, which favors TGF-α transcription in CAFs of the TME, which in turn promotes EGFR signaling, resulting in an increased metastatic ability of cancer cells. This mechanism generates a loop of epigenetic regulation between CAFs and cancer cells [88].

Other TME components that can have a role in regulating tumor cell expression pattern and development include immune cells, where TAMs were observed to increase DNMT1 expression in gastric cancer cells, leading to the silencing of the tumor suppressor gelsolin gene via hypermethylation [89].

Moreover, another epigenetically driven mechanism of communication between TME and tumor cells is the exosome release. In fact, both cancer and non-cancer cells in the TME secrete exosomes to “talk” with each other, and in these structures, ncRNAs can regulate metabolic reprogramming, angiogenesis, epithelial–mesenchymal transition, and ECM remodeling [90].

## 4. Conclusions

From this review, we can conclude that epigenetic mechanisms can have an important role in neuroendocrine tumor pathogenesis, representing interesting potential therapeutic targets but also prognostic and diagnostic biomarkers. For this reason, a further understanding of these regulatory mechanisms in neuroendocrine tumor cells is required in order to both define in more detail the impact of epigenetics on NEN pathogenesis and develop more efficient therapeutic strategies.

Another important player in tumoral progression is the tumor microenvironment, which, even through epigenetic mechanisms, is able to interact with cancer cells, promoting tumor progression, invasion, and immune response regulation. Consequently, it would be interesting to achieve a deeper knowledge on the complex molecular networks regulating the communication and balance between tumor cells and the surrounding microenvironment, especially considering the relatively limited knowledge about this topic in NETs. At present, the main gap consists of the mechanisms by which TME elements are able to influence cancer cells leading to tumoral progression, where epigenetic regulation can be a strong candidate, as suggested by other tumor types. Hence, further investigation is required to identify possible therapeutic targets that enable more efficient NEN treatment.

## Figures and Tables

**Figure 1 cancers-18-00069-f001:**
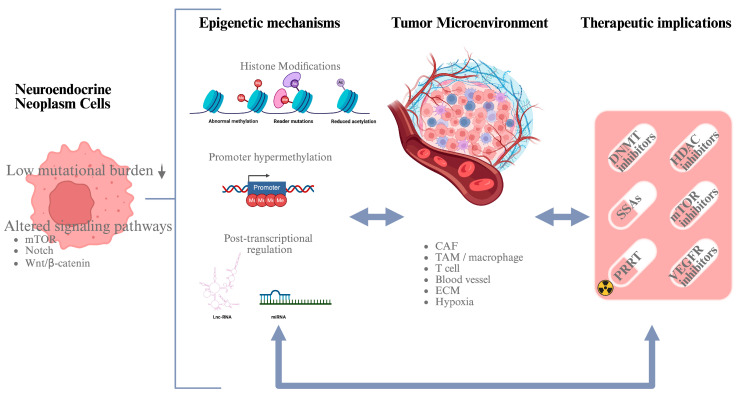
Schematic workflow illustrating the interplay between epigenetic mechanisms and the tumor microenvironment (TME) in neuroendocrine neoplasms (NENs). In NEN cells, characterized by a low mutational burden, epigenetic regulation plays a pivotal role; this includes DNA methylation, histone modifications, and noncoding RNA–mediated mechanisms. These epigenetic alterations are dynamically influenced by components of the TME, such as cancer-associated fibroblasts, immune cells, extracellular matrix, hypoxia, and extracellular vesicles, through bidirectional crosstalk. TME-driven epigenetic reprogramming contributes to tumor progression, therapy resistance, and immune evasion. Targeting epigenetic mechanisms in combination with established therapies—such as somatostatin analogs, peptide receptor radionuclide therapy, targeted agents, and cytotoxic chemotherapy—may overcome resistance mechanisms and improve therapeutic efficacy in NENs (created in https://BioRender.com).

**Figure 2 cancers-18-00069-f002:**
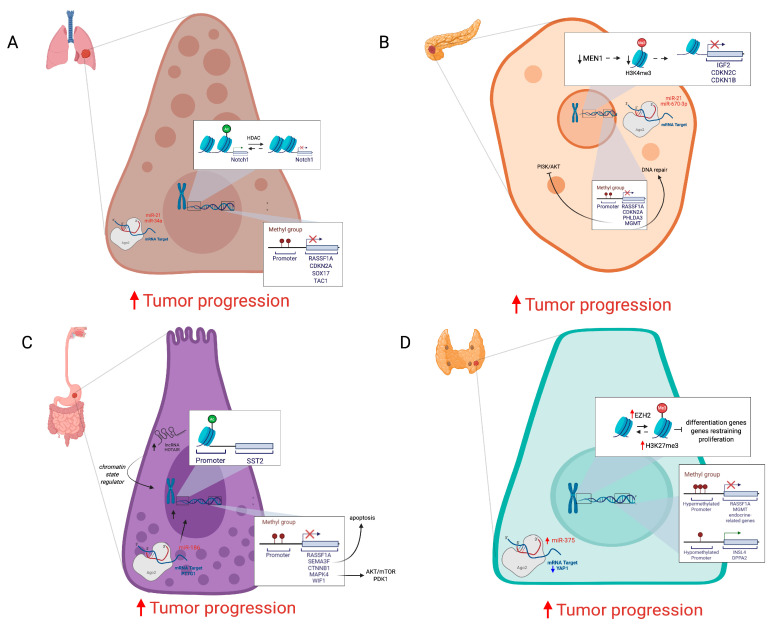
(**A**) Different types of altered epigenetic modifications resulting in favored tumor progression in lung NETs; (**B**) in pancreatic NETs; (**C**) in gastrointestinal NETs; (**D**) in thyroid NETs (created in https://BioRender.com).

**Table 1 cancers-18-00069-t001:** NET types, WHO subclassification, association with hereditary syndromes/genetic mutations.

Tumor Site	Origin Cells	Classification	Mutated Genes/Hereditary Syndromes
Lungs	Pulmonary neuroendocrine cells (neuroepithelial bodies)	Well-differentiated low-grade typical carcinoids (TCs) Well-differentiated intermediate grade atypical carcinoids (ACs) Poorly differentiated high-grade large-cell neuroendocrine carcinomas (LCNECs) Poorly-differentiated high-grade small-cell lung carcinomas (SCLCs) [5]	*MEN1*, *SWI*/*SNF* complex, *KMT2*/*MLL*, and *PSIP1* [6]
Pancreas	Islet cells of the pancreas	PanNECs Well-differentiated PanNETs [7]	*MEN1*, *VHL*, *NF-1*, tuberous sclerosis complex, and glucagon cell adenomatosis [8,9]
GI tract	Enterochromaffin cells of the gut neuroendocrine system	NET G1 (well-differentiated, low malignancy) NET G2 (well-differentiated, medium malignancy) NET G3 (well-differentiated, less aggressive than NECs) NECs (poorly differentiated very aggressive) [10]	*MEN1*, *VHL* syndrome, *NF1*, and tuberous sclerosis [11]
Thyroid	Mainly C-cells of the thyroid gland	Follicular cell-derived neoplasms Parafollicular cell-derived tumors [7]	MEN2 syndromes and *RET* [7]

## Data Availability

No new data were created or analyzed in this study.

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
