# Peer review of "Epigenetics and the Tumor Microenvironment in Neuroendocrine Tumors"

_cancers, 2025, doi:10.3390/cancers18010069_

Round 1
Reviewer 1 Report
Comments and Suggestions for Authors
The manuscript offers a detailed survey of current knowledge on epigenetic regulation and the role of the tumor microenvironment in neuroendocrine neoplasms. The topic is important and the review is generally well structured, but several sections would benefit from clearer explanations, stronger critical analysis, and a more cohesive synthesis of the cited work.
Comments
-
The link between specific epigenetic changes and their biological or therapeutic consequences in NETs needs to be developed more clearly. At present, the text describes individual findings without integrating them into a unified narrative.
• Which reported epigenetic alterations remain uncertain, conflicting, or poorly supported across published studies?
• Are there methodological issues in the cited literature—such as limited sample sizes or heterogeneous tumor grades—that might account for some of these inconsistencies? -
It would be helpful to include a conceptual framework or a simple model summarizing how epigenetic modifications might interact with, or be driven by, various components of the tumor microenvironment.
-
In the section discussing TME–epigenetic crosstalk, many examples are derived from non-NET tumors. The authors should point out which of these mechanisms are most likely to apply to NENs and justify why.
-
The brief reference to combined HDACi/DNMTi strategies does not fully explore the therapeutic implications.
• How do epigenetic patterns vary with tumor grade or proliferation index across NET subtypes?
• Are any epigenetic markers currently being explored clinically for diagnostic or prognostic use?
• Could the transfer of non-coding RNAs via extracellular vesicles serve as a practical biomarker or treatment target in NENs?
• What role might epigenetic alterations play in resistance to PRRT? -
The review focuses mainly on classical epigenetic processes. A short discussion of emerging mechanisms—such as RNA methylation (m6A), chromatin-remodeling complexes like SWI/SNF, or changes in higher-order chromatin architecture—would broaden the perspective and may be relevant to NET biology.
-
The conclusions would be strengthened by more explicitly outlining:
• the main unanswered questions in the field,
• key directions for future research, and
• how a deeper understanding of epigenetics and the TME might translate into improved clinical strategies for NENs.
Author Response
Reviewer#1
The manuscript offers a detailed survey of current knowledge on epigenetic regulation and the role of the tumor microenvironment in neuroendocrine neoplasms. The topic is important and the review is generally well structured, but several sections would benefit from clearer explanations, stronger critical analysis, and a more cohesive synthesis of the cited work.
We would like to thank the reviewer for the apreciation of our work, we have followed the suggestions and we believe we have significantly improved our manuscript.
Comments
- The link between specific epigenetic changes and their biological or therapeutic consequences in NETs needs to be developed more clearly. At present, the text describes individual findings without integrating them into a unified narrative.
• Which reported epigenetic alterations remain uncertain, conflicting, or poorly supported across published studies?
• Are there methodological issues in the cited literature—such as limited sample sizes or heterogeneous tumor grades—that might account for some of these inconsistencies?
We thank the reviewer for this important comment. In the revised manuscript, we have strengthened the integrative and critical discussion linking specific epigenetic alterations to their biological and therapeutic consequences in NETs.
It would be helpful to include a conceptual framework or a simple model summarizing how epigenetic modifications might interact with, or be driven by, various components of the tumor microenvironment.
In response to this suggestion, we have added a new conceptual figure (Figure 1) illustrating the bidirectional crosstalk between epigenetic mechanisms and the tumor microenvironment in neuroendocrine neoplasms.
In the section discussing TME–epigenetic crosstalk, many examples are derived from non-NET tumors. The authors should point out which of these mechanisms are most likely to apply to NENs and justify why.
We agree with the reviewer’s concern. In the revised version, we have expanded our discussion of NET-specific evidence and clearly distinguished data derived from neuroendocrine tumors from mechanisms extrapolated from other cancer types. We have incorporated recently published NET-specific studies, including hypoxia-driven epigenetic regulation in GEP-NETs, to strengthen the translational relevance of this section.
- The brief reference to combined HDACi/DNMTi strategies does not fully explore the therapeutic implications.
• How do epigenetic patterns vary with tumor grade or proliferation index across NET subtypes?
• Are any epigenetic markers currently being explored clinically for diagnostic or prognostic use?
• Could the transfer of non-coding RNAs via extracellular vesicles serve as a practical biomarker or treatment target in NENs?
• What role might epigenetic alterations play in resistance to PRRT?
We have substantially expanded this section to address the reviewer’s points. The revised manuscript now discusses how epigenetic patterns may vary across NET subtypes and grades, their relevance as diagnostic and prognostic biomarkers, and their potential role in resistance to SSAs and PRRT. We also include a dedicated discussion on extracellular vesicle–mediated transfer of non-coding RNAs as emerging biomarkers and therapeutic targets. These additions aim to provide a more nuanced and clinically relevant perspective on combination epigenetic strategies
- The review focuses mainly on classical epigenetic processes. A short discussion of emerging mechanisms—such as RNA methylation (m6A), chromatin-remodeling complexes like SWI/SNF, or changes in higher-order chromatin architecture—would broaden the perspective and may be relevant to NET biology.
As suggested, we have added a new subsection dedicated to emerging epigenetic mechanisms, including RNA methylation (m6A) and chromatin-remodeling complexes such as SWI/SNF. This section highlights their emerging relevance in neuroendocrine tumor biology and places them within the broader epigenetic landscape discussed in the review.
- The conclusions would be strengthened by more explicitly outlining:
• the main unanswered questions in the field,
• key directions for future research, and
• how a deeper understanding of epigenetics and the TME might translate into improved clinical strategies for NENs.
We have revised the Conclusions section to more clearly articulate the main unresolved questions in the field, key directions for future research, and the potential clinical implications of integrating epigenetic and TME-focused approaches in NEN management. This revised section emphasizes translational opportunities while acknowledging current limitations.
Reviewer 2 Report
Comments and Suggestions for Authors
The authors present a narrative review summarizing current knowledge on the role of epigenetic mechanisms, including DNA methylation, histone modifications, and non-coding RNAs, and their interplay with the tumor microenvironment in neuroendocrine neoplasms. The review synthesizes findings across major NEN subtypes, highlighting how epigenetic dysregulation may contribute to tumorigenesis, therapeutic resistance, and potential future treatment strategies targeting tumor-stroma interactions. Overall, the manuscript provides a broad overview of existing literature and aims to position epigenetics as a promising direction for research and therapy development in NENs.
Major comments: 1) The review currently lacks critical evaluation of the cited evidence; many statements are descriptive rather than analytical, and key controversies, gaps, or limitations in the literature are not discussed. 2) The manuscript mixes mechanistic details with high-level summaries without a consistent structure, resulting in sections that feel redundant or unfocused. 3) Several claims about therapeutic implications (e.g., combining epigenetic drugs with SSAs or TKIs) require more nuance, including discussion of clinical trial evidence, toxicity concerns, or translational feasibility. 4) In the introduction or discussion part, following reference about neuroendocrine tumors should be added to strengthen the paper: PMID: 30293109. 5) The manuscript would benefit from clearer integration between epigenetics and TME sections, as the link between these domains is proposed but not sufficiently supported by NEN-specific studies; much of the TME–epigenetics cross-talk content is extrapolated from other cancers. 6) Figures and tables require improved labeling and explanation; for example, Figure 1 summarizes epigenetic alterations but lacks adequate legend detail and specific citation mapping. 7) Definitions, abbreviations, and subtype terminology should be standardized throughout the text to improve readability. 8) Several sections are overly long and include fine-grained molecular details that may overwhelm readers; selective condensation and clearer thematic organization would improve flow. 9) The conclusions remain general and do not articulate concrete future research directions or specific hypotheses that emerge from the reviewed evidence.
Minor comments: Minor inconsistencies in formatting, spacing, and citation placement should be corrected for professionalism.
Author Response
The authors present a narrative review summarizing current knowledge on the role of epigenetic mechanisms, including DNA methylation, histone modifications, and non-coding RNAs, and their interplay with the tumor microenvironment in neuroendocrine neoplasms. The review synthesizes findings across major NEN subtypes, highlighting how epigenetic dysregulation may contribute to tumorigenesis, therapeutic resistance, and potential future treatment strategies targeting tumor-stroma interactions. Overall, the manuscript provides a broad overview of existing literature and aims to position epigenetics as a promising direction for research and therapy development in NENs.
We are grateful to the reviewer for their thoughtful and constructive comments. We have carefully considered all suggestions and revised the manuscript accordingly, resulting in a substantially improved version.
Major comments:
1) The review currently lacks critical evaluation of the cited evidence; many statements are descriptive rather than analytical, and key controversies, gaps, or limitations in the literature are not discussed.
We thank the reviewer for this important observation. In the revised manuscript, we have strengthened the critical evaluation of the available literature by explicitly discussing limitations such as small sample sizes, heterogeneity in tumor grade and anatomical site, and inconsistencies across studies. Where evidence remains incomplete or conflicting, this is now clearly highlighted, and key knowledge gaps are identified as priorities for future research,
2) The manuscript mixes mechanistic details with high-level summaries without a consistent structure, resulting in sections that feel redundant or unfocused.
We agree with this assessment and have substantially revised the manuscript structure to improve clarity and coherence.
3) Several claims about therapeutic implications (e.g., combining epigenetic drugs with SSAs or TKIs) require more nuance, including discussion of clinical trial evidence, toxicity concerns, or translational feasibility.
We have revised the therapeutic sections to provide a more nuanced discussion of epigenetic-based combination strategies. The revised manuscript now addresses translational feasibility, potential resistance mechanisms, and current limitations of clinical application, including variability in treatment response and the need for biomarker-driven patient selection.
4) In the introduction or discussion part, following reference about neuroendocrine tumors should be added to strengthen the paper: PMID: 30293109.
We thank the reviewer for this suggestion and have added the recommended reference to the revised manuscript to strengthen the background discussion on neuroendocrine tumor
5) The manuscript would benefit from clearer integration between epigenetics and TME sections, as the link between these domains is proposed but not sufficiently supported by NEN-specific studies; much of the TME–epigenetics cross-talk content is extrapolated from other cancers.
We agree with the reviewer and have improved the integration between epigenetics and tumor microenvironment sections. In particular, we now clearly distinguish NET-specific evidence from mechanisms extrapolated from other tumor types, providing explicit justification when the latter are discussed. Recently published NET-focused studies have been incorporated to reinforce this connection, and a conceptual framework summarizing this interplay has been added as a new figure (Figure 1)
6) Figures and tables require improved labeling and explanation; for example, Figure 1 summarizes epigenetic alterations but lacks adequate legend detail and specific citation mapping.
We have revised all figure legends to provide clearer explanations and improved labeling.
7) Definitions, abbreviations, and subtype terminology should be standardized throughout the text to improve readability.
We have carefully reviewed the entire manuscript to standardize terminology, abbreviations, and subtype definitions. An expanded abbreviations list has been added, and consistent nomenclature is now used throughout the text to improve readability and clarity.
8) Several sections are overly long and include fine-grained molecular details that may overwhelm readers; selective condensation and clearer thematic organization would improve flow.
In the revised manuscript, we have selectively condensed overly detailed sections while preserving key mechanistic insights relevant to NET biology. This restructuring improves flow and accessibility without compromising scientific accuracy.
9) The conclusions remain general and do not articulate concrete future research directions or specific hypotheses that emerge from the reviewed evidence.
We have substantially revised the Conclusions section to explicitly outline the main unanswered questions in the field, propose key directions for future research, and highlight how a deeper understanding of epigenetic regulation and tumor microenvironment interactions could translate into improved diagnostic and therapeutic strategies for neuroendocrine neoplasms.
Minor comments: Minor inconsistencies in formatting, spacing, and citation placement should be corrected for professionalism.
All formatting, spacing, and citation inconsistencies have been corrected in the revised manuscript.
Round 2
Reviewer 1 Report
Comments and Suggestions for Authors
All the required changes were done by the authors. I recommended publication of this manuscript.
Reviewer 2 Report
Comments and Suggestions for Authors
This manuscript has been revised according to the comments, it can be accepted for publication.